# ROBO-INSTRUCT: Simulator-Augmented Instruction Alignment For Finetuning Code LLMs

**Zichao Hu**
Department of Computer Science
UT Austin
Austin, TX 78712
zichao@utexas.edu

**Junyi Jessy Li**
Department of Linguistics
UT Austin
Austin, TX 78712
jessy@utexas.edu

**Arjun Guha**
Khoury College of Computer Sciences
Northeastern University
Boston, MA 02115
a.guha@northeastern.edu

**Joydeep Biswas**
Department of Computer Science
UT Austin
Austin, TX 78712
joydeepb@utexas.edu

## Abstract

Code LLMs have shown promising results with converting tasks in natural language to programs that can be executed by service robots. We are interested in finetuning small, specialized LLMs for this purpose, but collecting datasets of task-program pairs specific to each robot is time-consuming and expensive. While approaches such as SELF-INSTRUCT and EVOL-INSTRUCT are capable of generating novel tasks given a few examples, they are unable to provide the corresponding programs that correctly abide by physical-world and robot-constraints using the provided programming interface. Using a simulator is a natural potential solution to checking for such constraints, but building simulation environments that can handle arbitrary tasks and their necessary objects and locations, is challenging. To address these challenges, we introduce ROBO-INSTRUCT, which *synthesizes task-specific simulation environments on the fly* during program execution, by opportunistically inferring entity properties and enforcing corresponding constraints based on how the entities are used in the task program. Additionally, ROBO-INSTRUCT integrates an LLM-aided post-processing procedure to refine instructions for better alignment with robot programs. We demonstrate the effectiveness of ROBO-INSTRUCT across multiple LLMs, showing that our fine-tuned models outperform all baseline methods and even match or surpass the performance of several larger and proprietary models.

Project page: https://amrl.cs.utexas.edu/robo-instruct/

## 1 Introduction

Robot programs leverage robot skills, expressed as parameterized function calls, combined with common programming abstractions (loops, conditionals, etc) to perform complex open-world tasks. For example, by formulating robot manipulation and perception skills such as pick(object) and is_in_room(object), an LLM can generate a program for a service mobile robot to complete the task: *"Pick up an apple if you see one here."*. The state-of-the-art approaches in robotics use large, proprietary LLMs (*e.g.*, GPT) to generate such task-specific programs via in-context learning (Hu et al., 2024; Huang et al., 2023b; Biggie et al., 2023; Liu et al., 2023a; Wu et al., 2023; Liang et al., 2022; Singh et al., 2023; Huang et al., 2023a). While quite effective, such large models cannot be run locally on robots, require network connectivity to query remote LLM endpoints, increase response latency, and raise privacy concerns. Smaller models that can run locally on robots, on the other hand, are unfortunately

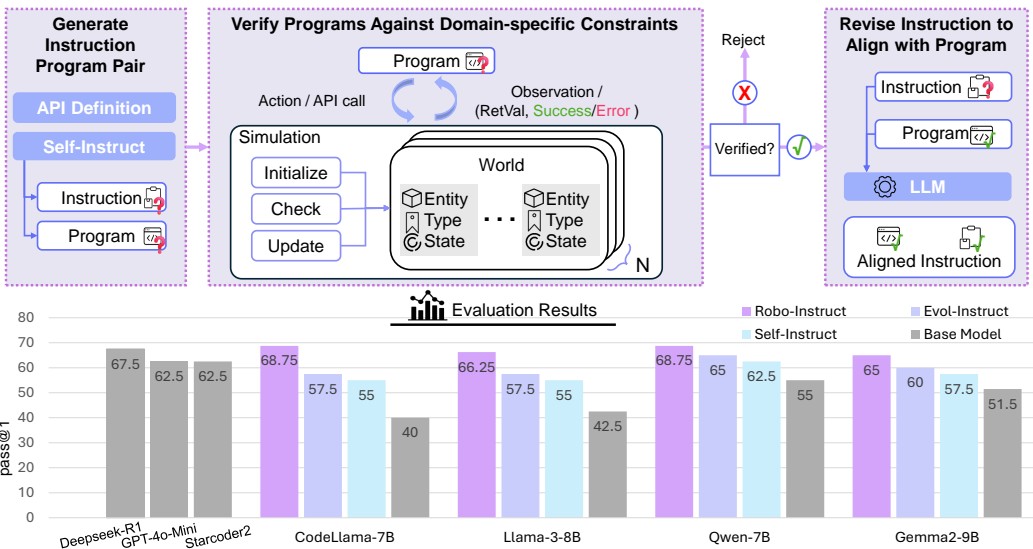

Figure 1: **High-level overview of the ROBO-INSTRUCT framework.** This figure also shows the pass@1 score performance of ROBO-INSTRUCT fine-tuned LLM compared to other LLMs on ROBOEVAL.

unable to match the performance of larger models — which naturally raises the question of how to finetune small models for robot-specific code generation.

Obtaining high-quality training data is crucial for fine-tuning LLMs. Unlike some domains where training data exist (*e.g.,* Meditron (Chen et al., 2023) for medical AI), robots vary in their capabilities, and manual data construction becomes unscalable. Thus, methods like SELF-INSTRUCT(Wang et al., 2022) and EVOL-INSTRUCT (Xu et al., 2024) provide promising approaches for generating synthetic training data.

However, generating robot programs presents unique challenges,

```
LLM Generated Program                                    1
def task_program():
 if not is_in_room("apple"):
  pick("apple")
```

**Real-world Constraint Violation**

Cannot pick up an apple that does not exist at the current location.

```
LLM Generated Program                                    2
def task_program():
 pick("apple")
 pick("apple")
```

**Robot Configuration Violation**

Cannot pick up two apples at once due to the robot having only one arm.

Figure 2: Examples of programs violating domain-specific constraints.

as robots interact with the real world and must adhere to robot- and environment-specific constraints, which these existing approaches do not verify. For instance, as illustrated in Fig. 2, a candidate program might instruct the robot to pick up an apple that is not present at its current location (example 1) or attempt to pick up multiple objects simultaneously, which is physically impossible if the robot can only hold one at a time (example 2). These constraints are domain-specific to the robot's intended tasks, and while a developer may recognize such violations, automating their detection is non-trivial. One potential solution is to execute the program in a robot simulator with well-defined environments. However, such simulations require pre-enumerating relevant entities and their states, which depend on the specific actions the program dictates. Since approaches like SELF-INSTRUCT generate a diverse range of programs, pre-enumerating all possible environments becomes impractical. Additionally, we observe that the generated instruction-program pairs may be inconsistent; Fig. 10 illustrates this problem, where the instruction specifies checking for an apple, but the program fails to perform this check.

To address these challenges, this work introduces ROBO-INSTRUCT, a framework for generating synthetic robot program training data to fine-tune open-weight LLMs for domain-

specific service robot tasks. As illustrated in Fig. 1, ROBO-INSTRUCT offers a principled approach for robot developers to define task constraints and verify candidate programs against such constraints. Drawing inspiration from Angelic Execution (Broy & Wirsing, 1981), ROBO-INSTRUCT opportunistically infers entity properties and enforces corresponding constraints by synthesizing simulation environments as the program executes. Once violations are detected, ROBO-INSTRUCT then employs a rejection sampling mechanism by invoking SELF-INSTRUCT to generate a new program based on the same instruction. To further address misalignment between the candidate instruction and program, ROBO-INSTRUCT incorporates an LLM-aided post-processing step that refines the instruction to better reflect the verified program's intent.

We show the effectiveness of ROBO-INSTRUCT by fine-tuning several LLMs to generate domain-specific robot programs, and evaluating them using ROBOEVAL (Hu et al., 2024), a benchmark designed for service mobile robots. Our ROBO-INSTRUCT-fine-tuned models significantly outperform their corresponding base models, achieving an average 19.9% improvement in pass@1 scores. They also outperform their SELF-INSTRUCT-fine-tuned and EVOL-INSTRUCT-fine-tuned counterparts by 9.7% and 7.2%, respectively. Moreover, the ROBO-INSTRUCT-fine-tuned models surpass or match the performance of several larger code models, including GPT-4o-mini (OpenAI et al., 2024a), Starcoder2-15B (Lozhkov et al., 2024), and Deepseek-R1-Qwen-32B (DeepSeek-AI et al., 2025).

# 2 ROBO-INSTRUCT

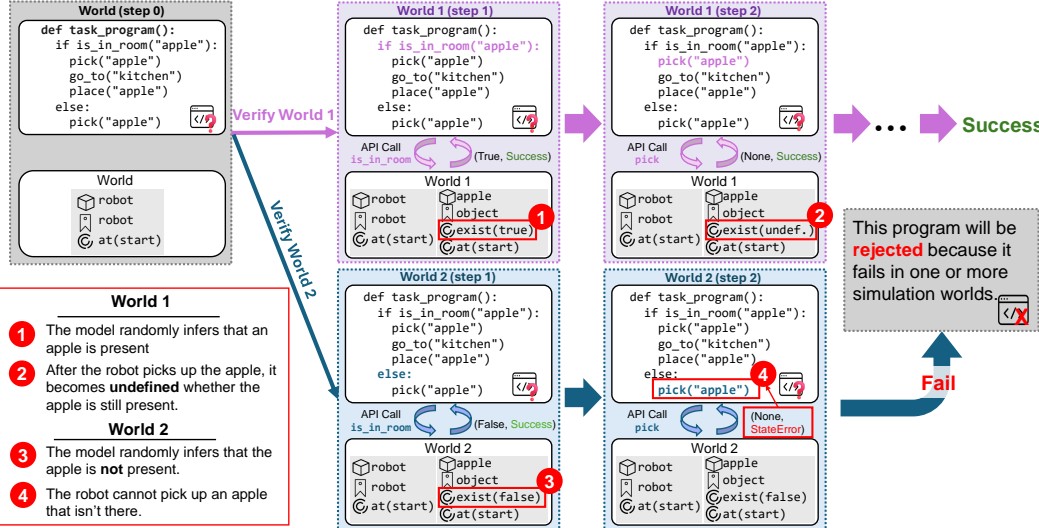

Figure 3: Illustration of ROBO-INSTRUCT executing a task program while incrementally building the simulation environment. The environment starts with only the robot's initial position (gray, step 0). As the program runs, it branches into two possible execution paths. To evaluate each path, two simulation environments are sampled (world 1 and world 2). In this example, the program fails because it attempts to pick up an apple that isn't present.

## 2.1 Overall Framework

ROBO-INSTRUCT generates task and robot program pairs as training data to fine-tune open-weight LLMs for domain-specific service robot tasks. As shown in Fig. 1, ROBO-INSTRUCT first uses SELF-INSTRUCT to propose novel tasks. For each task, using in-context learning, it prompts a LLM to generate a candidate program to perform the task using the robot APIs in the given context (detailed prompts in Appendix A.5.2). Then ROBO-INSTRUCT verifies the candidate program by synthesizing a simulation environment *on-the-fly* as API functions are executed (explained in Sec. 2.2). When the simulator catches violations of

domain-specific constraints, it rejects the candidate program and re-prompts the LLM for a new candidate program. If the program successfully terminates with no simulation failures, ROBO-INSTRUCT synthesizes additional simulation environments (up to a pre-defined limit) to check for the correctness of the candidate program from different initial configurations and environmental states. ROBO-INSTRUCT is thus able to catch candidate programs that are not robust to environmental variations. Finally, once the candidate program is verified, ROBO-INSTRUCT incorporates an LLM-assisted instruction-program alignment procedure, which revises natural language instructions using the verified candidate programs to enhance alignment between the two (as detailed in Sec. 2.3). Fig. 3 shows an example of how ROBO-INSTRUCT executes and verifies a candidate program while incrementally constructing the simulation environments (world 1 and world 2) on-the-fly. In the following sections, we present these components in detail.

## 2.2 Verifying Candidate Programs Against Domain-specific Constraints

To verify candidate programs, we introduce an algorithm inspired by Angelic Execution (Broy & Wirsing, 1981), which infers program properties from incomplete API specs. As shown in Pseudocode 1, it lets developers combine task constraints with robot APIs, automatically synthesizing simulation environments and detecting constraint violations during execution. The algorithm is built around three core concepts essential for service robots to reason about:

1. Different *entities*, e.g., "apple", "kitchen".

2. The *type* of the entities, and hence their affordances, e.g., "apple" is an object, you can pick it up; "kitchen" is a location, you can go to it, and it contains objects.

3. The *state* of the entities in the world, e.g., the "apple" is in the "kitchen".

---

**Pseudocode 1** ROBO-INSTRUCT for Simulating Robot API calls.

```
1  def RoboInstruct_Sim (api_function_call, args):
2    world = GetCurrentSimulationEnvironment()
3    entity_names = InferEntityNames(api_function_call, args)
4    for entity_name in entity_names:
5      if EntityIsNew(world, entity_name):
6        # Synthesize a new entity inferred from the API call
7        required_type = InferRequiredType(api_function_call, args)
8        entity = InitEntityWithEmptyState(world, entity_name, required_type)
9        # The entity's state can be initialized either randomly or deterministically, based
         ↪  on the API function and its arguments
10       inferred_state = InferRandomOrDeterministicState(entity_name,
11                                                          api_function_call, args)
12     else:
13       # Check if the entity's type and state is consistent with the API call
14       required_type = InferRequiredType(api_function_call, args)
15       inferred_type = GetEntityType(world, entity_name)
16       if required_type != inferred_type:
17         return [None, 'TypeError']
18       state_requirements = GetStateRequirements(world, entity_name)
19       inferred_state = InferNextState(entity_name, api_function_call, args)
20       if not IsStateConsistent(inferred_state, state_requirements):
21         return [None, 'StateInconsistentError']
22     # Update the state of the entity in the world
23     world = UpdateWorldState(world, entity_name, inferred_state)
24   # Randomly sample a value consistent with API call and the current world
25   ret_val = InferReturnVal(world, api_function_call, args)
26   return [ret_val, 'Success']
```

---

These concepts are closely tied to the robot APIs, where each API invocation during program execution updates the simulation environment. For example, the go_to(loc) action takes

only entities of type "location" as arguments, and executing it changes the *state* of the robot to be at the new location.

Unlike static simulation environments, ROBO-INSTRUCT synthesizes the simulation environment *on-the-fly*. Consider an API function call like is_in_room(obj), which takes an argument of type "object" and checks whether that object is present in the robot's current location. When this API is called with a specific parameter (*e.g.*, is_in_room("apple")), ROBO-INSTRUCT performs the following steps, as illustrated in Pseudocode 1. First, it retrieves the current simulation environment and infers the entity name (*e.g.*, "apple") (lines 2–3). Next, it checks whether an entity with this name has already been initialized (line 6). If not, ROBO-INSTRUCT synthesizes the entity by inferring its type and state based on the API call (line 7-11).

A feature of ROBO-INSTRUCT is that an entity's state can be initialized randomly or deterministically, depending on the specific API function (line 10). For example, consider two API function calls: is_in_room("apple") and go_to("kitchen"). is_in_room("apple") randomly initializes the state of the "apple", as the "apple" may or may not be present in the same room as the robot. In contrast, go_to(kitchen") deterministically updates the robot's location to the "kitchen" (assuming the API is executed successfully). This flexibility allows ROBO-INSTRUCT to simulate diverse environments and test program robustness under varying conditions.

If an entity has already been initialized during program execution, its type and state are checked against domain-specific constraints (lines 14–21). First, the entity's type must remain consistent across different API calls (lines 14–17). For example, consider the program shown in Program Example 1. The API pick(obj) expects an argument of type object, while go_to(loc) expects a location [1]. By executing API calls sequentially, ROBO-INSTRUCT first infers that "apple" is an object. Then when go_to("apple") is called, ROBO-INSTRUCT detects a type inconsistency and returns with an error (line 17). Next, ROBO-INSTRUCT computes the requirements for the robot's next state based on the current world state and compares them with the inferred next state (lines 18–21). If the inferred state violates these requirements, ROBO-INSTRUCT raises an error (line 21). As illustrated in bullet point 4 of Fig. 3, ROBO-INSTRUCT detects a constraint violation when the robot attempts to pick up an "apple" that does not exist in the environment. This mismatch between the expected and actual state leads ROBO-INSTRUCT to reject the candidate program.

```
def task_program():
    pick("apple")
    go_to("apple")
```

**Program Example 1**

Another feature of ROBO-INSTRUCT is that the states of entities in the simulator resemble STRIPS-style planning (Fikes & Nilsson, 1971), where each state can be either "true" or "false", as illustrated in bullet points 1 and 3 of Fig. 3. However, unlike traditional STRIPS planning, ROBO-INSTRUCT also explicitly includes an "undefined" value for states. This value represents the default state of any entity not explicitly defined during a program's execution. For instance, as shown in bullet point 2 of Fig. 3, after the robot picks up an apple, ROBO-INSTRUCT marks the apple's state as "undefined" since it does not track how many apples remain in the environment and cannot determine whether an apple still exists at the robot's location. As a result, this state information is omitted in subsequent executions (a more detailed comparison with STRIPS planning is discussed in Appendix A.2).

Finally, ROBO-INSTRUCT infers return values according to the API specification and the current simulation context. As shown in bullet points 1 and 3 of Fig. 3, ROBO-INSTRUCT randomly decides whether the apple is present, which leads to different return values across simulation runs. A program is considered valid if it terminates successfully in all simulated environments.

---

[1] In this example, type compatibility check is strict (i.e., "apple" is only an object and no further inference is made about its location). Nevertheless, the algorithm is also capable of handling more advanced scenarios.

### 2.3 LLM-aided Instruction-Program Alignment Procedure

A key challenge in generating synthetic training data is the mismatch between instructions and programs—*e.g.,* a program may skip a step implied by the instruction (see Appendix Fig. 10). Rejection sampling only checks program validity, not whether it fully matches the instruction. Thus, valid programs can still fail to fulfill the instruction's intent.

To address this challenge, ROBO-INSTRUCT employs a post-processing procedure to align instructions with their corresponding programs. The key intuition behind this approach is that since the program has already been verified, it can remain fixed, and the *task shifts to finding an instruction that accurately aligns with the program.* Hence, leveraging the advanced code understanding capabilities of modern LLMs (Rozière et al., 2024; Nam et al., 2024; Leinonen et al., 2023; Li et al., 2023; Lekshmi-Narayanan et al., 2024), ROBO-INSTRUCT applies Chain-of-Thought reasoning (Wei et al., 2022) to generate and compare revised instructions, selecting the one that best reflects the program. Detailed prompt designs for this process are provided in AppendixA.5.4.

## 3 Analysis and Experiments

### 3.1 Experiments Setup

**Benchmark.** In this work, we evaluate on the ROBOEVAL benchmark (Hu et al., 2024), a domain-specific program generation benchmark for service mobile robots (See Appendix A.3 for details.). In this domain, a service mobile robot can perceive objects, navigate to various locations, manipulate items, and communicate with humans. Accordingly, we design ROBO-INSTRUCT to align with the constraints of the APIs used in this benchmark. We use the pass@1 metric to assess the performance of LLMs in generating correct robot programs.

**Data Generation.** To generate a diverse dataset, we choose to use the open-weight Llama-3 model (Grattafiori et al., 2024) with nucleus sampling to create instruction-program pairs, setting the temperature $T = 1$ and the top $p = 0.95$. The maximum resampling limit is capped at 3 to accommodate instructions that initially produce invalid programs, and each verification process duplicates the program 100 times to ensure robust probabilistic coverage of different execution branches. For the LLM used for post-processing, we empirically adjust the generation temperature to $T = 0.3$ to optimize performance (See Fig. 7). Furthermore, we assess the edit similarity between token sequences of each instruction pair in the dataset (Lee et al., 2022), removing duplicates where the similarity score exceeds 0.6. The same similarity-based approach is used to decontaminate the dataset against the ROBOEVAL benchmark tasks (more details are presented in Appendix A.4.3).

**Training Setup.** We used PEFT (Hu et al., 2022) with unsloth (Unslothai, 2024) to fine-tune four popular open-weight CodeLLMs, including Codellama-Python (Rozière et al., 2024), Llama3 (Grattafiori et al., 2024), Qwen2.5-Coder (Hui et al., 2024), and Gemma2(Team et al., 2024). The learning rate is set to be 3*e*-5 with a warmup ratio of 3% and a constant lr scheduler. We employ the AdamW optimizer (Loshchilov & Hutter, 2019) with an effective batch size of 8, training each model for 5 epochs using a sequence length of 2048 tokens.

**Baselines.** We compare the performance of the ROBO-INSTRUCT fine-tuned models against the same models fine-tuned using two popular data generation methods: SELF-INSTRUCT(Wang et al., 2022) and EVOL-INSTRUCT(Xu et al., 2024). Additionally, we compare their performance against larger models, categorized into two groups: (1) proprietary LLMs, including GPT (OpenAI et al., 2024b); and (2) open-weight LLMs, including Codellama-Python-34B, Starcoder2-15B (Lozhkov et al., 2024), and Deepseek-R1-Distill-Qwen-32B (DeepSeek-AI et al., 2025).

### 3.2 Is ROBO-INSTRUCT Effective at Generating Training Data to Fine-Tune a Small Language Model for Generating Domain-Specific Robot Programs?

Tab. 1 presents the average pass@1 results for different LLMs on ROBOEVAL using two decoding settings: greedy decoding ($T = 0$) and nucleus sampling ($T = 0.2$). ROBO-INSTRUCT-fine-tuned models outperform base models by an average of 19.9% in pass@1

| Fine-tune | Model | # Param | ROBOEVAL pass@1 | | Licensing |
| | | | $T = 0$ | $T = 0.2$ | |
|---|---|---|---|---|---|
| - | GPT-4.5 | - | 88.75% | 88.25% | Proprietary |
| - | GPT-4 | - | 83.75% | 85.81% | Proprietary |
| - | GPT-4o-mini | - | 62.50% | 61.63% | Proprietary |
| - | Codellama-Python | 34B | 46.25% | 48.25% | Open |
| - | Starcoder2 | 15B | 62.5% | 60.94% | Open |
| - | DeepSeek-R1-Qwen | 32B | 67.50% | 65.13% | Open |
| - | Codellama-Python | 7B | 40.00% | 39.31% | Open |
| Self-Instruct | CodeLlama-Python | 7B | 55.00% | 52.69% | Open |
| Evol-Instruct | CodeLlama-Python | 7B | 57.50% | 55.38% | Open |
| Robo-Instruct (ours) | CodeLlama-Python | 7B | **68.75%** | **66.00%** | Open |
| - | Llama3 | 8B | 42.5% | 36.69% | Open |
| Self-Instruct | Llama3 | 8B | 55.00% | 53.75% | Open |
| Evol-Instruct | Llama3 | 8B | 57.50% | 54.87% | Open |
| Robo-Instruct (ours) | Llama3 | 8B | **66.25%** | **62.44%** | Open |
| - | Qwen2.5-Coder | 7B | 55.00% | 55.25% | Open |
| Self-Instruct | Qwen2.5-Coder | 7B | 62.50% | 59.38% | Open |
| Evol-Instruct | Qwen2.5-Coder | 7B | 65.00% | 62.75% | Open |
| Robo-Instruct (ours) | Qwen2.5-Coder | 7B | **68.75%** | **67.00%** | Open |
| - | Gemma2 | 9B | 51.50% | 52.00% | Open |
| Self-Instruct | Gemma2 | 9B | 57.50% | 57.88% | Open |
| Evol-Instruct | Gemma2 | 9B | 60.00% | 59.50% | Open |
| Robo-Instruct (ours) | Gemma2 | 9B | **65.00%** | **62.63%** | Open |

Table 1: Pass@1 results of different LLMs on ROBOEVAL computed with greedy decoding $T = 0$ and nucleus sampling $T = 0.2$.

scores and surpass their SELF-INSTRUCT-fine-tuned and EVOL-INSTRUCT-fine-tuned counterparts by 9.7% and 7.2%, respectively (analyses of the generated data in Appendix A.4.3). Notably, despite having significantly fewer parameters, ROBO-INSTRUCT-fine-tuned models match or exceed the performance of larger open-weight models and even the proprietary GPT-4o-mini.

### 3.3 Evaluating the Contributions of ROBO-INSTRUCT Components

| Method | T=0 | | T=0.2 | | Invalid Programs |
| | pass@1 | Improv. | pass@1 | Improv. | |
|---|---|---|---|---|---|
| Codellama-7B-Python | 40.00% | +0% | 39.31% | +0% | 38.31% |
| SELF-INSTRUCT | 55.00% | +15.00% | 52.69% | +13.38% | 20.94% |
| +Reject Unsolvable (RU) | 60.00% | +20.00% | 57.62% | +18.31% | 23.38% |
| +Verify Program + RU | 63.75% | +23.75% | 63.88% | +24.57% | **14.13%** |
| +LLM-aided Align + RU | 58.75% | +18.75% | 59.81% | +20.50% | 23.44% |
| +Both (ROBO-INSTRUCT) | **68.75%** | **+28.75%** | **66.00%** | **+26.69%** | 17.07% |

Table 2: Pass@1 results of different methods on ROBOEVAL computed with greedy decoding $T = 0$ and nucleus sampling $T = 0.2$. The **Invalid Programs** column indicates the percentage of programs that result in execution errors when tested on ROBOEVAL tasks.

We conduct an ablation study to examine how verifying programs against domain-specific constraints (+Verify Program) and applying the LLM-aided Instruction-Program Alignment procedure (+LLM-aided Align) affect the performance of ROBO-INSTRUCT. Since SELF-INSTRUCT may generate instructions for which no corresponding valid program can be generated given an instruction, we include Reject Unsolvable (RU) as an additional baseline. SELF-INSTRUCT+RU keeps only the instructions that lead to at least one successful program

execution and removes those that do not produce any valid results. Tab. 2 shows the average pass@1 results from CodeLlama-7B-Python fine-tuned on different datasets generated by each method. Results from SELF-INSTRUCT + RU indicate that simply discarding invalid instructions improves model performance. In addition, using either "Verify Program" or "LLM-aided Align" alone improves upon the baseline SELF-INSTRUCT results, and incorporating both within ROBO-INSTRUCT achieves the best pass@1 performance. For more ablation experiments and analysis on the generated data, we refer the readers to Appendix A.1 for more results.of

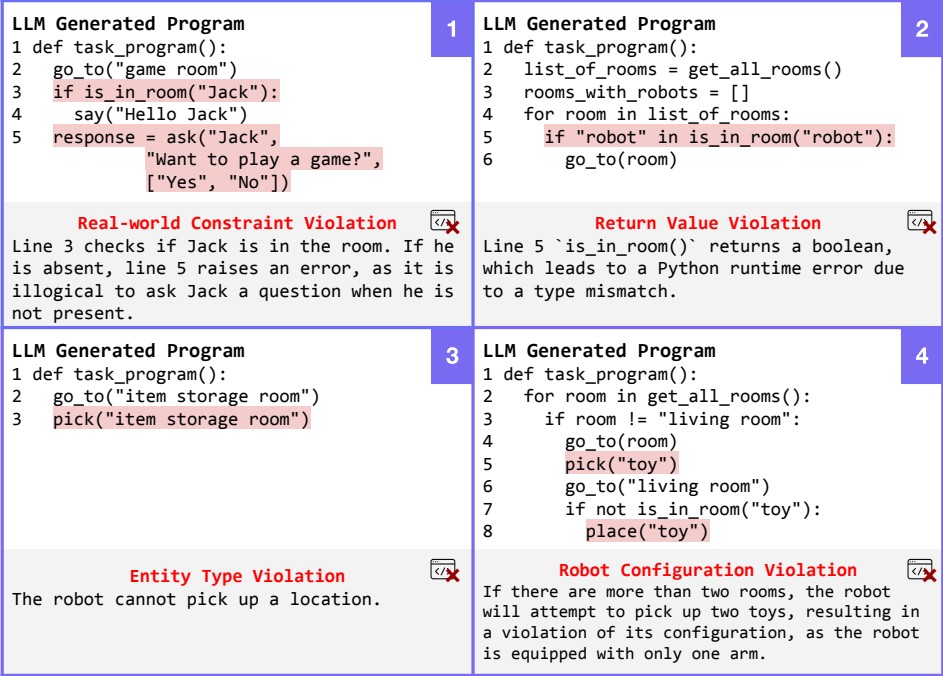

Figure 4: SELF-INSTRUCT-Generated Program Errors. Examples highlight errors that violate domain-specific constraints.[2]

## 3.4 Qualitative analysis of the generated program errors

We qualitatively analyze invalid programs identified by ROBO-INSTRUCT, as shown in Fig. 4. The first three examples are easily recognizable to humans as flawed. However, the last example is more complex and involves an error when the robot can navigate to more than two rooms. After the robot places a toy in the living room, ROBO-INSTRUCT updates the environment to reflect that a toy is now in the room (line 8). However, when the robot returns to the living room later (line 6), it will not drop the item it's holding (line 8). As a result, when the robot enters a third room (line 4) and tries to pick up another toy (line 5), an error will occur because the robot is only capable of carrying one item at a time. This example demonstrates that ROBO-INSTRUCT can detect invalid programs beyond those easily identifiable through human inspection.

## 3.5 Real-World Deployment Demo

We deployed the ROBO-INSTRUCT fine-tuned model (on a local 3080 Ti and an H100 server) to generate and execute mobile robot programs in real-world environments, as illustrated in Fig. 5. Unlike GPT models, our locally deployed fine-tuned model offers significantly

---

[2]Programs have been adapted to succinctly demonstrate the types of errors and fit within the figure.

| Models | GPT-4.5 | GPT-4 | GPT-4o-Mini | Robo-Instruct (Local) | Robo-Instruct (Server) |
|---|---|---|---|---|---|
| **Inference Speed** | 10 tokens/s | 19 tokens/s | 41 tokens/s | **57 tokens/s** | **114 tokens/s** |

Table 3: Inference speed of different models.

faster program generation. Additional results on long-horizon tasks beyond ROBOEVAL are presented in Appendix A.6.

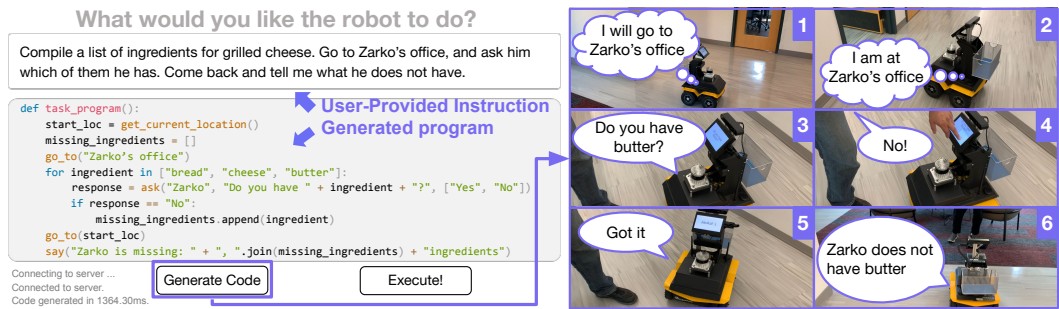

Figure 5: Deployment of the ROBO-INSTRUCT fine-tuned model to generate programs based on user-provided instructions and execute them on the robot.

## 4 Related Work

**LLMs for Robot Code Generation** LLMs are performant in generating robot programs from natural language (Liang et al., 2022; Singh et al., 2023; Huang et al., 2023a). One common approach involves generating composable costmaps for motion planning, as seen in Voxposer (Huang et al., 2023b) for tabletop tasks and NavCon (Biggie et al., 2023) for navigation. LLMs are also effective at creating reward functions—Eureka (Ma et al., 2023; 2024) and Language-to-Rewards (Yu et al., 2023) enable robots to learn complex skills through LLM-generated rewards. For high-level planning, LLM+p (Liu et al., 2023a) outputs PDDL plans, while Tidybot (Wu et al., 2023) learns user preferences from examples to generate sequential task programs. RoboEval (Hu et al., 2024) targets service robots, generating and validating domain-specific programs for long-horizon tasks.

**Generating Datasets For Fine-tuning LLMs** To improve code generation, many studies create specialized datasets (Muennighoff et al., 2024; Köpf et al., 2023; Muennighoff et al., 2022). SELF-INSTRUCT(Wang et al., 2022) is a popular approach that uses LLMs to generate synthetic data. This approach is later extended by Code Alpaca (Chaudhary, 2023) and Gorilla-LM (Patil et al., 2023) for code and ML APIs. In addition, Evol-Instruct (Xu et al., 2024; Luo et al., 2024) proposes an approach to iteratively update instructions to become more complex through different prompting strategies. OSS-Instruct (Wei et al., 2023) uses open-source code to train Magicoder, matching GPT-3.5-Turbo on HumanEval (Chen et al., 2021). While prior work focuses on seed instruction generation, we explore post-processing methods, especially for robotics programs (Hu et al., 2024), where we can effectively leverage constraints to filter out erroneous programs.

**Relevance to Program Analysis** Our approach is inspired by angelic execution (Broy & Wirsing, 1981), where we apply nondeterminism to resolve the types and state of input arguments for each predefined robot API, in the context of LLM-based program generation. Outside the application of LLMs, related ideas have been explored in program analysis techniques such as symbolic execution, exemplified by KLEE (Cadar et al., 2008), which generates high-coverage tests for complex, environment-intensive programs. Large-scale static analysis tools (Calcagno et al., 2015; Bessey et al., 2010; Ayewah et al., 2008) also demonstrate the effectiveness of analyzing codebases at scale to uncover bugs and enforce correctness properties.

## 5   Conclusion, Limitation and Future Works

In this work, we introduce ROBO-INSTRUCT, a novel framework for generating synthetic robot program training data to fine-tune open-weight LLMs for domain-specific service robot tasks. ROBO-INSTRUCT features a novel algorithm to synthesize simulation environments on-the-fly to check for any violations of domain-specific constraints and an LLM-aided instruction alignment procedure that refines instructions to better match the generated programs. Experimental results show that ROBO-INSTRUCT-fine-tuned models significantly outperform baseline approaches using SELF-INSTRUCT and EVOL-INSTRUCT, while also matching or surpassing larger open-weight LLMs and proprietary models like GPT-4o-mini in generating service robot programs. However, ROBO-INSTRUCT is not without limitations. *The framework enforces necessary—but not sufficient—conditions for program correctness*: while a program that fails our checks is guaranteed to violate at least one domain-specific constraint, a passing program is not necessarily correct in all possible scenarios. For instance, consider a simplified program containing only the instruction *pick_up("building")*. ROBO-INSTRUCT may synthesize a scenario in which a "building" is treated as a pickable object, which is clearly unrealistic. As a result, while ROBO-INSTRUCT effectively detects many domain-specific violations, it may miss feasibility issues beyond the defined task constraints. Despite this limitation, our experiments demonstrate that ROBO-INSTRUCT is effective in identifying a wide range of domain-specific violations. Future work could explore integrating ROBO-INSTRUCT within a reinforcement learning fine-tuning loop, allowing models to iteratively learn from violations, thereby improving their ability to generate robust and realistic robot programs for domain-specific applications.

## Acknowledgments

This work was partly supported by NSF grants CCF-2313027, IIS-2416461, CAREER-2046955 and UT Austin through the Associate Professor Experimental (APX) program.

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

# A Appendix

## A.1 Overview

In this appendix, we first outline the relationship between ROBO-INSTRUCT and the classic STRIPS planning formulation in subsection A.2, providing a new perspective on the proposed algorithm. Subsection A.3 shows mode detailed descriptions of the ROBOEVAL benchmark. In subsection A.4, we present additional ablation experiments to analyze the percentage of invalid programs generated by SELF-INSTRUCT and the effectiveness of the rejection-sampling strategy combined with ROBO-INSTRUCT. We also explore how the generation temperature in the LLM-aided Instruction-Program Alignment Procedure impacts final performance and compare the dataset diversity produced by ROBO-INSTRUCT and SELF-INSTRUCT. Subsection A.5 lists the seed tasks used in ROBOEVAL and the CoT prompts. in subsection A.6, we report real-world experiments that empirically evaluate the performance of our fine-tuned model on two long-horizon tasks, which differ significantly from those in ROBOEVAL, and assess the model's latency in generating programs. Although this work focuses on service mobile robots, the proposed framework is adaptable to other domains. In subsection A.7, we offer toy examples showing how ROBO-INSTRUCT can be extended to verify programs by incorporating domain-specific constraints.

## A.2 Relevance to STRIPS planning

The proposed ROBO-INSTRUCT shares significant similarities with the formulation of STRIPS planning. A STRIPS instance is typically represented as a tuple $\langle I, G, A, P \rangle$, where $I$ denotes the initial state of the simulation environment, $G$ represents the desired goal state that the robot aims to achieve, $A$ defines the set of actions available to transition between states, and $P$ is the set of preconditions that must be satisfied before performing actions. Thus, ROBO-INSTRUCT can be reformulated to align with the STRIPS formulation as shown in Alg. 2. Each API invocation corresponds to an action, and its precondition consists of a set of literals, representing specific combinations of entities, types, and states.

To address this, we extend the classic STRIPS formulation by incorporating dynamically discovered literals. Unlike the conventional STRIPS approach, where each literal is binary—True when defined and False when not—we introduce a third value, "Undefined." This means a literal must be explicitly defined as either True or False; otherwise, it remains in the Undefined state. When an action requires a literal that is undefined, a random value (True or False) is assigned to it, and the literal is added to the state of the simulation environment (line 7). Once the precondition is fully defined, the action is executed, and domain-specific constraints are checked for any violations (line 10). This extension enables ROBO-INSTRUCT to handle arbitrary programs effectively.

---

**Pseudocode 2** ROBO-INSTRUCT − STRIPS(api_fn, params, $\mathcal{W}$)

---

1: **Input:** api_fn                                                                  ▷ The API function name
2: **Input:** api_inputs                                                  ▷ The input received by the API invocation
3: **Input:** $\mathcal{W}$                                         ▷ The current state of the simulation environment
4: $p \leftarrow$ GETPRECOND(api_fn, params)              ▷ Get the parameter-specific precondition for api_fn
5: **for** $l \in p$ **do**                                         ▷ Loop through every literal in the precondition
6:   **if** CHECKDEFINED($\mathcal{W}, l$) **is** Undefined **then**
7:     $\mathcal{W} \leftarrow$ GROWWORLD($l, \mathcal{W}$)      ▷ Randomly instantiate the literal and grow $\mathcal{W}$ to include it
8:   **end if**
9: **end for**
10: retval, $\mathcal{W} \leftarrow$ EXECUPDATE(api_fn, params, $\mathcal{W}$)                    ▷ Execute api_fn and update $\mathcal{W}$
11: **return** retval

---

## A.3 ROBOEVAL Benchmark

ROBOEVAL is a domain-specific code generation benchmark, featuring a suite of 16 tasks designed to evaluate the ability of LLMs to understand custom APIs and generate programs for service robots. In this domain, a service robot can perceive objects, navigate to various

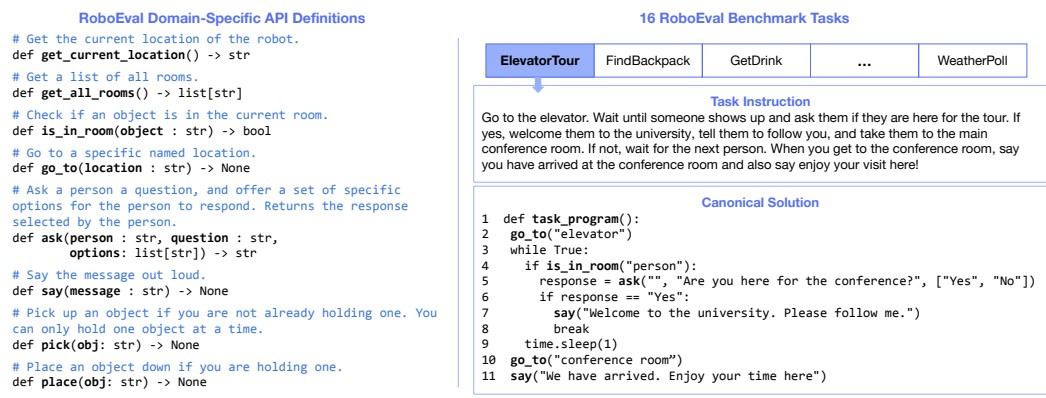

Figure 6: ROBOEVAL APIs and benchmark task example.

locations, manipulate items, and communicate with humans. Furthermore, the robot should be capable of basic commonsense reasoning and executing complex tasks that involve conditional and repetitive actions. To facilitate these capabilities, ROBOEVAL defines a set of 8 API functions in Python as skill primitives. Fig. 6 illustrates these function signatures and definitions, alongside an example task instruction and its canonical solution from the benchmark. In addition, unlike other popular code generation benchmark tasks (Chen et al., 2021; Austin et al., 2021; Li et al., 2022; Liu et al., 2023b; Lai et al., 2022; Hendrycks et al., 2021), *the order of the robot's actions is crucial for successfully completing the specified tasks*. For instance, in the task *"bring me a marker from the classroom that does not have a whiteboard,"* the robot must check each classroom until it finds one without a whiteboard, whereas simply bringing back a marker is insufficient. Hence, ROBOEVAL evaluates the generated program by executing it in a simulator to capture the action traces, which are subsequently validated for sequence correctness using temporal logic.

## A.4 Ablation Experiments

### A.4.1 the Effectiveness of the Rejection-Sampling Strategy

We analyze the percentage of instruction-program pairs discarded by ROBO-INSTRUCT at various maximum resampling limits, as shown in Fig. 7. Initially, with the maximum resampling limit set to 0, disabling the rejection-sampling method, approximately 51% of the programs generated by SELF-INSTRUCT contain errors. As the limit increases, fewer programs are discarded. However, there is a diminishing return; even with the maximum resampling limit set to 10, about 15% of the instructions still result in invalid programs.

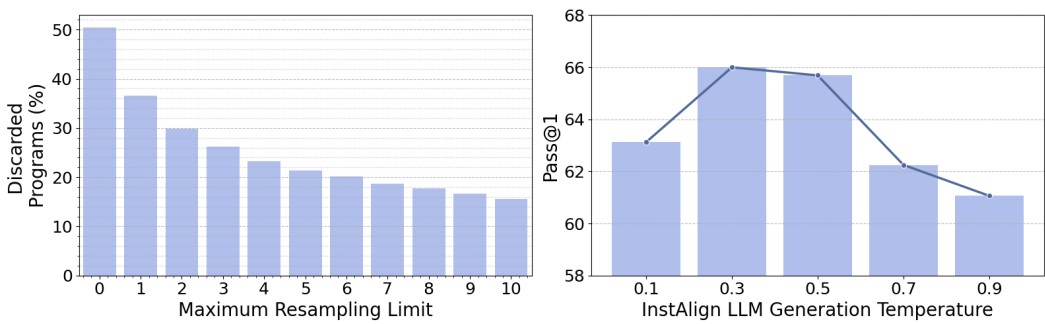

Figure 7: Ablation Experiment Results

### A.4.2 *Instruction Alignment model temperature*

We further investigate how varying LLM temperatures for generating the revised instruction in the LLM-aided Instruction-Program Alignment Procedure impact the performance of the fine-tuned model. Fig. 7 shows the bar chart of the pass@1 score of the models fine-tuned over datasets generated using different LLM temperatures. The model performs the best when fine-tuned on the dataset generated using LLM temperature $T = 0.3$. As the temperature increases, we observe a decrease in performance.

### A.4.3 *Analysis of Generated Dataset*

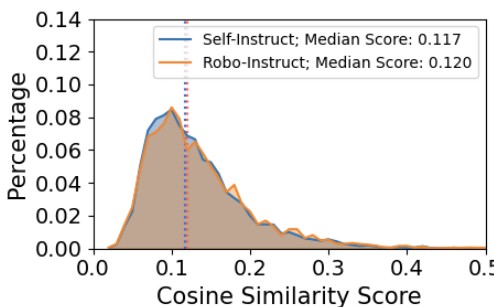

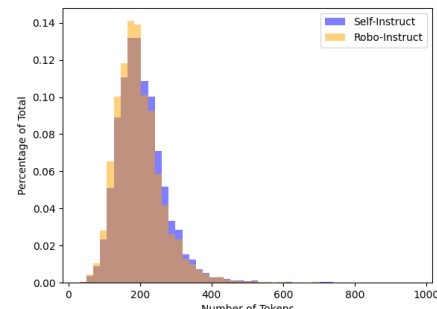

Figure 8: Cosine similarities between ROBOE-VAL and generated data.

Figure 9: Token length distribution for each instruction-program pair.

Similar to Magicoder (Wei et al., 2023), we show the improvements from ROBO-INSTRUCT are not merely due to selection bias, i.e., including data more aligned with the distribution of ROBOEVAL tasks than SELF-INSTRUCT. We pair each sample from the generated dataset with task instructions and their canonical solutions, then compute cosine similarity using TF-IDF embeddings (Sparck Jones, 1988). Fig. 8 shows comparable token similarities between both methods. Additionally, Fig. 9 presents the token length distribution, which also appears similar for both.

| Method | Size | Ngram=4 Score | # Synth. Loc. | # Synth. Obj. |
|---|---|---|---|---|
| SELF-INSTRUCT | 5K | 0.581 | 956 | 1060 |
| ROBO-INSTRUCT | 5K | 0.587 | 1025 | 928 |

Table 4: Dataset Statistics

Since ROBO-INSTRUCT does not rely on pre-defined simulation environments, we aim to assess the diversity of programs generated by SELF-INSTRUCT and whether ROBO-INSTRUCT can maintain this diversity. To do so, we measure the number of distinct entities, such as synthetic locations and objects. As shown in Tab. 4, with a dataset of only 5,000 samples, approximately 1,000 unique objects and locations are generated, highlighting that conventional robot simulations with pre-defined environments are insufficient. Additionally, Tab. 4 presents the n-gram diversity scores for each dataset, indicating that both distributions and dataset statistics are highly similar. This suggests that ROBO-INSTRUCT not only preserves but enhances the quality of generated data compared to SELF-INSTRUCT, rather than simply aligning the dataset with benchmark tasks.

### A.5 Prompts

### A.5.1 ROBOEVAL *Seed Task Example*

Seed Task Example 1:

```
1   # Instruction: Go to Arjun's office, ask him if he is ready to head out,
2   # and come back and tell me what he said
3   def task_program():
4       start_loc = get_current_location()
5       go_to("Arjun's office")
6       response = ask("Arjun",
7           "Are you ready to go?",
8           ["Yes", "No"])
9       go_to(start_loc)
10      say("Arjun said: " + response)
```

Seed Task Example 2:

```
1   # Instruction: Ask Alice if she needs 1, 2, or 3 boxes.
2   # Go to the storage room and ask if they have that many boxes.
3   # If so, go place the boxes in Alice's office.
4   # Otherwise, tell Alice you could not get the boxes.
5   def task_program():
6       go_to("Alice's office")
7       num_boxes = ask("Alice",
8           "How many boxes do you need?",
9           ["1", "2", "3"])
10      go_to("storage room")
11      response = ask("",
12          "Do you have" + num_boxes + " boxes?",
13          ["Yes", "No"])
14      if response == "Yes":
15          for _ in range(int(num_boxes)):
16              pick("box")
17              go_to("Alice's office")
18              place("box")
19              go_to("storage room")
20      else:
21          go_to("Alice's office")
22          say("I could not get the boxes")
```

Seed Task Example 3:

```
1   # Instruction: Check if there is a red marker in the main
2   # office, and if so, tell Eve that there is a marker there.
3   # If not, go to the supply room and
4   # bring a red marker to the main office.
5   def task_program():
6       go_to("main office")
7       red_marker_found = is_in_room("red marker")
8       if red_marker_found:
9           go_to("Eve's office")
10          say("There is a red marker in the main office")
11      else:
12          go_to("supply room")
13          pick("red marker")
14          go_to("main office")
15          place("red marker")
```

Seed Task Example 4:

```
1   # Instruction: Check every classroom if there is a whiteboard.
2   # Go to Aiden's office to tell him which room does not
```

```
3    # have a whiteboard. Come back and tell me task is completed.
4    def task_program():
5        start_loc = get_current_location()
6        list_of_rooms = get_all_rooms()
7        room_without_whiteboard = []
8        for room in list_of_rooms:
9            if "classroom" not in room:
10               continue
11           go_to(room)
12           if not is_in_room("whiteboard"):
13               room_without_whiteboard.append(room)
14       go_to("Aiden's office")
15       if len(room_without_whiteboard) > 0:
16           message = ""
17           for room in room_without_whiteboard:
18               message += room + ", "
19           message += "do not have a whiteboard"
20       else:
21           message = "all classrooms have a whiteboard"
22       say(message)
23       go_to(start_loc)
24       say("task is completed")
```

Seed Task Example 5:

```
1    # Instruction: Go to the kitchen and wait for someone
2    # to show up. When someone shows up, ask them to open
3    # the fridge, then pick up a diet coke.
4    # Finally, put the diet coke in the living room.
5    def task_program():
6        go_to("kitchen")
7        while True:
8            if is_in_room("person"):
9                response = ask("",
10                   "Please open the fridge",
11                   ["Yes", "No"])
12               if response == "Yes":
13                   pick("diet coke")
14                   break
15           time.sleep(1)
16       go_to("living room")
17       place("diet coke")
```

Seed Task Example 6:

```
1    # Instruction: Take a bed sheet from the laundry room
2    # and put it in each of the bedrooms.
3    def task_program():
4        start_loc = get_current_location()
5        list_of_rooms = get_all_rooms()
6        for room in list_of_rooms:
7            if "bedroom" not in room:
8                continue
9            go_to("laundry room")
10           pick("bed sheet")
11           go_to(room)
12           place("bed sheet")
13       go_to(start_loc)
```

### A.5.2  *Prompts to Generate Synthetic Dataset Using* SELF-INSTRUCT

```
1   You are a helpful assistant. Here is a robot that has the
2   following capabilities:
3   - def get_current_location() -> str:
4   - def get_all_rooms() -> list[str]:
5   - def is_in_room(object : str) -> bool:
6   - def go_to(location : str) -> None:
7   - def ask(person : str, question : str, options: list[str]) -> str:
8   - def say(message : str) -> None:
9   - def pick(obj: str) -> None:
10  - def place(obj: str) -> None:
11  Generate an interesting robot task that can be accomplished using the
12  above capabilities.
13  {SEED EXAMPLE 1}
14
15  ...
16
17  Generate an interesting robot task that can be accomplished using the
18  above capabilities.
19  {SEED EXAMPLE 6}
20
21  Generate an interesting robot task that can be accomplished using the
22  above capabilities.
```

### A.5.3  *LLM-aided Instruction-Program Alignment Procedure*

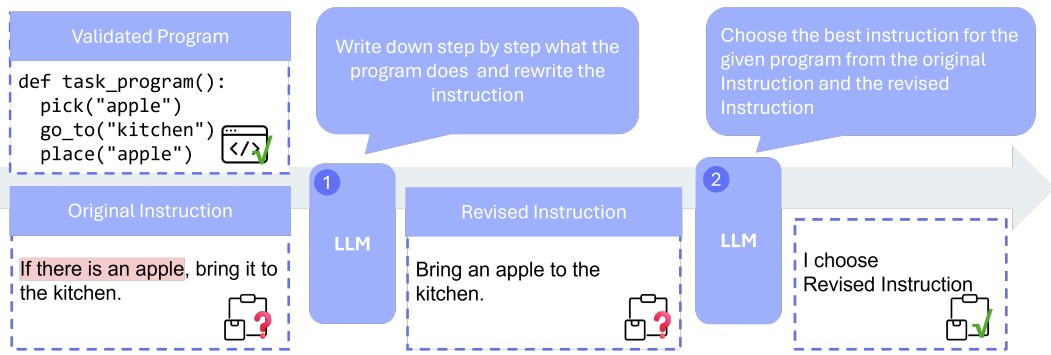

Figure 10: Overview of the LLM-aided Instruction-Program Alignment Procedure.

### A.5.4  *CoT Prompts for LLM-aided Instruction-Program Alignment Procedure*

```
1   ### Role
2   You are an expert at understanding robot programs.
3   You will be given a task instruction and robot program pair.
4   However, the instruction may not align with the program well.
5   You need to correct the task instruction to match the given robot program.
6
7   ### Context
8   The robot only has access to the following 8 APIs and
9   standard Python functions
10  - def get_current_location() -> str:
11  - def get_all_rooms() -> list[str]:
12  - def is_in_room(object : str) -> bool:
13  - def go_to(location : str) -> None:
14  - ask(person : str, question : str, options: list[str]) -> str:
15  - say(message : str) -> None:
```

```
16  - def pick(obj: str) -> None:
17  - def place(obj: str) -> None:
18
19  ### Inputs
20  Original Instruction
21      This is a task instruction that may not align with the robot program
22  Robot Program
23      This is a python function starting with `def task_program():`
24
25  ### Task
26  1. Write down all the provided APIs used in the program and
27  explain the effect of each API in this program
28  2. Examine these APIs and write down step by step what the program does
29  3. Combine all the results above and rewrite the instruction
30  You need to be specific and clear in your final corrected instruction.
```

## A.6  Real-world Experiments

### A.6.1  *Program Generation Results Beyond* RoboEval

We conducted a small qualitative experiment to assess the performance of the base model, SELF-INSTRUCT fine-tuned models, ROBO-INSTRUCT fine-tuned models, and GPT-4 on long-horizon tasks. To showcase performance, we created two new long-horizon tasks that differ significantly from those in the ROBOEVAL benchmark. Using a temperature setting of 0, we generated programs with each model and found that our ROBO-INSTRUCT fine-tuned model successfully produced correct programs, while both the base model and SELF-INSTRUCT fine-tuned model failed. Additionally, GPT-4 made an error on the second instruction. We show the generation results from ROBO-INSTRUCT fine-tuned models and GPT-4 below.

**Task instruction 1:**

```
1  Let's play a game: Double and give it to the next person. Start with
2  1 dollar. Go to rooms A, B, C, D, E, F, and G. If you see someone, tell
3  them how much money you have. Then ask if they would like to take the
4  money now or double the amount and give it to the next person. If they
5  choose to take it, the game is over, and you should come back to me.
6  Otherwise, double your money and continue. If, in the end, no one takes
7  the money, tell me how much you still have.
```

**ROBO-INSTRUCT fine-tuned model (Ours) [Success]**

```python
1  def task_program():
2      start_loc = get_current_location()
3      list_of_rooms = ["A", "B", "C", "D", "E", "F", "G"]
4      money = 1
5      for room in list_of_rooms:
6          go_to(room)
7          if is_in_room("person"):
8              say("I have $" + str(money))
9              response = ask("", """Would you like to take the money now
10             or double the amount and give it to the next person?""",
11             ["Take now", "Double and give"])
12             if response == "Take now":
13                 go_to(start_loc)
14                 say("The game is over, I gave the money to " + room)
15                 return
16             else:
17                 money *= 2
```

```
18        go_to(start_loc)
19        say("No one took the money, I still have $" + str(money))
```

### GPT-4 [Success]

```
1   def task_program():
2       start_loc = get_current_location()
3       rooms = ["A", "B", "C", "D", "E", "F", "G"]
4       money = 1
5       for room in rooms:
6           go_to(room)
7           if is_in_room("person"):
8               say("I have " + str(money) + " dollars")
9               response = ask("", """Would you like to take the money now
10              or double the amount and give it to the next person?""",
11              ["Take it", "Double it"])
12              if response == "Take it":
13                  say("Game over. The person in room " + room +
14                  " took the money.")
15                  go_to(start_loc)
16                  return
17              else:
18                  money *= 2
19       go_to(start_loc)
20       say("No one took the money. I still have " + str(money) + " dollars.")
```

### Task instruction 2:

```
1   Go to my office and check if I have a table, a chair, and a monitor
2   there. If any of these items are missing, go to Jason's office and
3   see if he is there. If he is, ask him if I can borrow the missing items.
4   If he agrees, pick up each missing item and bring it to my office.
5   If Jason is not in his office or he says no, come back and tell me the
6   reason.
```

### ROBO-INSTRUCT fine-tuned model (Ours) [Success]

```
1   def task_program():
2       start_loc = get_current_location()
3       go_to("my office")
4       table_found = is_in_room("table")
5       chair_found = is_in_room("chair")
6       monitor_found = is_in_room("monitor")
7       if not table_found:
8           go_to("Jason's office")
9           if is_in_room("Jason"):
10              response = ask("Jason", "Can I borrow the table from you?",
11              ["Yes", "No"])
12              if response == "Yes":
13                  go_to("Jason's office")
14                  pick("table")
15                  go_to("my office")
16                  place("table")
17              else:
18                  go_to(start_loc)
19                  say("Jason said no")
20          else:
21              go_to(start_loc)
22              say("Jason is not in his office")
23       if not chair_found:
```

```
24              go_to("Jason's office")
25              if is_in_room("Jason"):
26                  response = ask("Jason", "Can I borrow the chair from you?",
27                  ["Yes", "No"])
28                  if response == "Yes":
29                      go_to("Jason's office")
30                      pick("chair")
31                      go_to("my office")
32                      place("chair")
33                  else:
34                      go_to(start_loc)
35                      say("Jason said no")
36              else:
37                  go_to(start_loc)
38                  say("Jason is not in his office")
39          if not monitor_found:
40              go_to("Jason's office")
41              if is_in_room("Jason"):
42                  response = ask("Jason", "Can I borrow the monitor from you?",
43                  ["Yes", "No"])
44                  if response == "Yes":
45                      go_to("Jason's office")
46                      pick("monitor")
47                      go_to("my office")
48                      place("monitor")
49                  else:
50                      go_to(start_loc)
51                      say("Jason said no")
52              else:
53                  go_to(start_loc)
54                  say("Jason is not in his office")
55          go_to(start_loc)
56          say("task is completed")
```

### GPT-4 [Failure]

```
1   def task_program():
2       start_loc = get_current_location()
3       go_to("my office")
4       missing_items = []
5       for item in ["table", "chair", "monitor"]:
6           if not is_in_room(item):
7               missing_items.append(item)
8       if len(missing_items) > 0:
9           go_to("Jason's office")
10          if is_in_room("Jason"):
11              for item in missing_items:
12                  response = ask("Jason", "Can I borrow your " + item +
13                  "?", ["Yes", "No"])
14                  if response == "Yes":
15  # BUG: did not return to Jason's office after going back to my office
16                      pick(item)
17                      go_to("my office")
18                      place(item)
19                  else:
20                      go_to(start_loc)
21                      say("Jason did not allow me to borrow his " + item)
22                      return
23          else:
24              go_to(start_loc)
25              say("Jason is not in his office")
26              return
```

```
27      go_to(start_loc)
28      say("Task is completed")
```

### A.6.2 Generation Latency

**What would you like the robot to do?**

Go to the elevator. Wait until someone shows up and ask them if they are here for the tour. If yes, welcome them to the university, tell them to follow you, and take them to the main conference room. If not, wait for the next person. When you get to the conference room, say you have arrived at the conference room and also say enjoy your visit here!

```python
def task_program():
    start_loc = get_current_location()
    while True:
        go_to("elevator")
        if is_in_room("person"):
            response = ask("", "Are you here for the tour?", ["Yes", "No"])
            if response == "Yes":
                say("Welcome to the university! Please follow me.")
                go_to("main conference room")
                break
        time.sleep(1)
    say("I have arrived at the conference room. Enjoy your visit here!")
    go_to(start_loc)
```

[ Execute! ]          [ Generate Code ]

Connecting to server...
Connected to server. Generating code...
Code generated in 1162.00ms.          ⬅  Generated by our model

(a) Example Program 1 generated by our fine-tuned model.

**What would you like the robot to do?**

Go to the elevator. Wait until someone shows up and ask them if they are here for the tour. If yes, welcome them to the university, tell them to follow you, and take them to the main conference room. If not, wait for the next person. When you get to the conference room, say you have arrived at the conference room and also say enjoy your visit here!

```python
def task_program():
    go_to("elevator")
    while True:
        if is_in_room("person"):
            response = ask("", "Are you here for the tour?", ["Yes", "No"])
            if response == "Yes":
                say("Welcome to the university! Please follow me to the main conference room.")
                go_to("main conference room")
                say("We have arrived at the main conference room. Enjoy your visit here!")
                break
        time.sleep(1)
```

[ Execute! ]          [ Generate Code ]

Connecting to server...
Connected to server. Generating code...
Code generated in 6822.60ms.          ⬅  Generated by GPT-4

(b) Example Program 1 generated by GPT-4.

Figure 11: Our fine-tuned model is approximately 6x faster in inference speed than GPT-4 (Part 1).

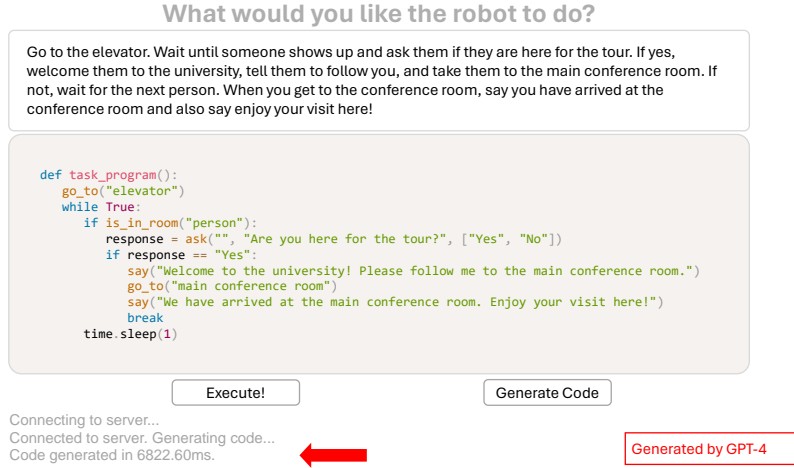

(c) Another Example Program generated by GPT-4.

(d) Another Example Program generated by GPT-4.

Figure 12: Our fine-tuned model is approximately 6x faster in inference speed than GPT-4 (Part 2).

## A.7 Toy Examples Beyond Service Mobile Robots

### A.7.1 Robot with low-level controls

Consider a tabletop manipulation scenario with a potential API function, is_rotate(robot_gripper_ name, radians), where the robot's gripper has a physical constraint, allowing rotation only within the range $\left[-\frac{\pi}{6}, \frac{\pi}{6}\right]$ radians. For the following generated program snippet:

```
def task_program():
    rotate("left hand", math.pi/6)
    rotate("left hand", math.pi/6)
    rotate("left hand", math.pi/6)
```

ROBO-INSTRUCT will first infer that "left hand" is an entity of the robot gripper type. Then, regardless of the initial configuration of the gripper, ROBO-INSTRUCT will throw an error because the program causes the gripper to exceed its allowable range of motion.

### A.7.2 AI-powered personal digital assistant

Consider a broader application than robotics: code generation for an AI-powered personal digital assistant. This AI assistant could handle scheduling events using an API function like schedule_on_calendar(event, start_time, duration). Given the instruction: *"My schedule is free tomorrow morning. Please create two 1-hour timeslots for office hours for my robotics and deep learning class."* The assistant could generate a program to create these timeslots:

```python
def task_program():
    schedule_on_calendar("robotics class office hour",
                         "9:30 am", "1 hr")
    schedule_on_calendar("deep learning class office hour",
                         "10:00 am", "1 hr")
```

In this example, ROBO-INSTRUCT needs to reason about the entities "robotics class office hour" and "deep learning class office hour", which are categorized as event types. The event type indicates that these entities have associated timeslots. The state of these entities is defined by the time they occur: robotics class office hour is set for 9:30-10:30 am, and deep learning class office hour is set for 10:00-11:00 am. During evaluation, ROBO-INSTRUCT can identify a time conflict between these two office hours and thus determine that the generated program is invalid.

