# OpenReview forum: "Robo-Instruct: Simulator-Augmented Instruction Alignment For Finetuning Code LLMs"
_colmweb.org/COLM/2025/Conference — COLM 2025_

### Official Review · Reviewer_txhX · 2025-04-30

**Rating:** 6
**Confidence:** 3
**Ethics Flag:** 1

**Summary:**

The paper proposes a synthetic data generation method that generates a diverse set of (natural language instruction, robot program) pairs from a starting seed set. The method is based on self-instruct and uses what seems like a dynamic program analysis method to verify the validity of generated robot programs and discards invalid programs. Finally, the valid remaining programs are checked for alignment with the natural language instruction, and the instruction is rewritten to align with the program. This gives a high quality dataset for tuning smaller, specialized models that can be deployed on a robot.

**Questions To Authors:**

- There is a vast literature on program analysis that seems relevant to your work. For example, see "KLEE: Unassisted and Automatic Generation of High-Coverage Tests for Complex Systems Programs" or "A few billion lines of code later: using static analysis to find bugs in the real world". Can you kindly cite some relevant program analysis papers since it seems connected to your approach?

**Reasons To Accept:**

- The paper communicates the method clearly, is well-written, and is a useful addition to the toolbox of synthetic data generation methods.

- Introduces program analysis methods to the language model community.

**Reasons To Reject:**

- The application of program analysis to synthesized programs to synthetic data generation is sound, but I'm unsure of whether there is anything novel in the program analysis method itself. Still, this might of interest to the language model research community who might not be aware of the body of work in program analysis. It would be good to cite relevant work from venues such as PLDI, OSDI, SOSP, so that interested readers can follow-up.

- As the results in the paper show, many of the frontier models are outperforming the models finetuned on the proposed synthetic data generation method. So, the value of the proposed method in forward looking terms is not so clear. Would it make sense to show results on distillation from larger models and compare that with your method? Of course, distillation from a larger, often closed model is not always an option, but it would be good to know how your method holds up.

- Not forward looking: Language models have improved significantly over the past 2 years, and evidence is mounting that late stage task-specific finetuning does not help as much as before and might even harm generalization. So, would it make sense apply your program verification methods directly in the RL stage? i.e., prompt the model with the task and use your method to provide a reward based on the validity of the generated program.

---

> ### Author Response · Authors · 2025-06-01
> **Response 2**
>
> > The application of program analysis to synthesized programs to synthetic data generation is sound, but I'm unsure of whether there is anything novel in the program analysis method itself. Still, this might of interest to the language model research community who might not be aware of the body of work in program analysis. It would be good to cite relevant work from venues such as PLDI, OSDI, SOSP, so that interested readers can follow-up.
>
> We would like to stress that the goal of this paper is to tackle a key challenge in testing robot programs for correctness: unlike software programs or coding problems, robot programs act like “policies” whose correctness depends on satisfying domain-specific constraints. To our knowledge, there is no prior approach that can handle this verification challenge for arbitrary robotics tasks, especially when working with a broad set of robot APIs. While the idea of angelic execution (from CAAP'81, which we cited) has inspired recent work in program verification, it has not been applied to LLM training or to verifying programs in embodied AI. Our work is the first to extend this technique to embodied AI, where inference involves reasoning about physical-world entities rather than just program states, making it a novel and valuable contribution. To ensure interested readers can follow up, we will expand the related work section to explain how our work fits within the broader landscape of program verification and analysis, and to cover the closest works in the PL community,
>
> > There is a vast literature on program analysis that seems relevant to your work. For example, see "KLEE: Unassisted and Automatic Generation of High-Coverage Tests for Complex Systems Programs" or "A few billion lines of code later: using static analysis to find bugs in the real world". Can you kindly cite some relevant program analysis papers since it seems connected to your approach?
>
> Thank you for the suggestions! As mentioned in our previous response, we will expand our review of related work and include the suggested references (along with other relevant works) to ensure that interested readers can follow up and explore these areas in greater depth. However, we would also like to note that in this work, we are primarily inspired by Angelic Execution, which we have cited in our text. While works such as “KLEE: Unassisted and Automatic Generation of High-Coverage Tests for Complex Systems Programs” and “A Few Billion Lines of Code Later: Using Static Analysis to Find Bugs in the Real World” are foundational in program analysis, they are not directly applicable to our setting as we focus on testing and verifying robot programs that must respect embodiment constraints in the physical world.

---

> > ### Comment · Reviewer_txhX · 2025-06-02
> >
> > Sounds good! Any references to relevant constraint solvers would be great too. (ex: would the cassowary constraint solver be relevant?)

---

> ### Author Response · Authors · 2025-06-01
> **Response 1**
>
> We appreciate the reviewer's constructive feedback. We will provide our responses below.
>
> > As the results in the paper show, many of the frontier models are outperforming the models finetuned on the proposed synthetic data generation method. So, the value of the proposed method in forward looking terms is not so clear. Would it make sense to show results on distillation from larger models and compare that with your method? Of course, distillation from a larger, often closed model is not always an option, but it would be good to know how your method holds up.
>
> As we noted in the Introduction, our primary focus is on models that can run on-device, such as those deployed in robots, where continuous internet access is not always available. While it’s true that the largest proprietary models can outperform our proposed method, there is significant interest in the community to improve the capabilities of smaller, edge-deployable models that can operate with limited compute resources [1,2,3]. Regarding the suggestion to distill from larger models, we performed a preliminary evaluation using GPT-4 to distill LLaMA-3 models, but only achieved around 60% pass@1 rate—lagging behind our finetuned LLaMA-3 model’s 66.25%.
>
> > Not forward looking: Language models have improved significantly over the past 2 years, and evidence is mounting that late stage task-specific finetuning does not help as much as before and might even harm generalization. So, would it make sense apply your program verification methods directly in the RL stage? i.e., prompt the model with the task and use your method to provide a reward based on the validity of the generated program.
>
> We agree that integrating RoboInstruct in an RL loop is a promising future direction, and we are actively pursuing this line of work. However, at the time of this study, supervised finetuning was still a dominant paradigm (e.g., Magicoder [4], WizardCoder [5], CodeAlpaca [6], GorillaLM [7], etc), and our work focuses on identifying and filtering out programs that violate domain-specific constraints for robotics applications—a novel and orthogonal contribution. While incorporating RoboInstruct into RL is an exciting avenue, our proposed method is orthogonal to the SFT vs. RL debate and provides valuable contributions to the field in its own right.
>
>
> **References:**
> 1. Chen et al. FASTNav: Fine-tuned Adaptive Small-Language-Models Trained for Multi-point Robot Navigation
> 2. Sikorski et al. Deployment of Large Language Models to Control Mobile Robots at the Edge
> 3. Xu et al. On-Device Language Models: A Comprehensive Review
> 4. Wei et al. Magicoder: Empowering Code Generation with OSS-Instruct
> 5. Luo et al. WizardCoder: Empowering Code Large Language Models with Evol-Instruct
> 6. Chaudhary et al. Code Alpaca: An Instruction-following LLaMA Model trained on code generation instructions
> 7. Patil et al. Gorilla: Large Language Model Connected with Massive APIs

---

### Official Review · Reviewer_LLwZ · 2025-05-07

**Rating:** 7
**Confidence:** 3
**Ethics Flag:** 1

**Summary:**

This paper describes a method to generate a dataset of programs in the service robotics domain from from natural language instructions. Using an existing large language model (LLM), the authors create a superset of initial instructions--program pairs. They automatically check the programs are valid using a simulator. The key idea of the simulation is that the authors infer properties and constraints on the objects and physical environment of the simulation. They can then reject generated programs that could not execute in the physical world. The authors then use the remaining programs to fine-tune instruction-based large language models. They also revise the text of the instructions--program pairs to align better the textual instructions with the working programs. The authors carried out an evaluation of fine-tuned models of open weight LLMs and a comparison with two other generators of instructions--program pairs. They report improvements over the other systems.

**Questions To Authors:**

Will you release your code?

**Reasons To Accept:**

* The idea is interesting and seems new
* The description is very pedagogical.
* The appendices contain many valuable details
* The authors used clear figures to explain the principles of their method.
* The evaluation on the RoboEval dataset is thorough and shows an improved performance over Self-Instruct and Evol-Instruct

**Reasons To Reject:**

* The task examples are a bit simplistic and it would be interesting to discuss how the system performs with more complex cases
* The description of the limitations is quite sketchy, which makes it difficult to assess what the system can really do beyond RoboEval
* There is no analysis of nonworking programs. It would help the reader understand the limitations of the system
* The authors did not commit to releasing their code. Although the appendix provides details, it would be rather difficult to reproduce this system

---

> ### Author Response · Authors · 2025-06-01
> **Response**
>
> We appreciate the reviewer's constructive feedback and we will address their comments below.
>
> > The task examples are a bit simplistic and it would be interesting to discuss how the system performs with more complex cases
>
> The task examples in Figure 4 were simplified for clarity of presentation, with the goal of highlighting prominent cases where programs violate domain-specific constraints. In practice, the actual generated instruction programs can be more complex. In addition, Appendix A6.1 shows examples of these longer and more intricate tasks, while Appendix A7 demonstrates how RoboEval can be applied to other robotics applications with different APIs. Upon acceptance, we will release both the raw generated dataset and the RoboEval-verified dataset.
>
> > The description of the limitations is quite sketchy, which makes it difficult to assess what the system can really do beyond RoboEval
>
> Thank you for pointing this out. We can expand on the limitations section to be more explicit. To clarify, RoboInstruct checks necessary—but not sufficient—conditions for program correctness: if a program fails our checks, it is guaranteed to be incorrect in at least one scenario, but a passing program is not guaranteed to be universally correct. This stems from our use of "angelic execution," which instantiates entities based on their use in the code. For example, in the case of the program:
> ```python
> if (is_in_room("building")):
>     pick_up("building")
>     go_to("mail room")
>     put_down("building")
> ```
> RoboInstruct may synthesize a scenario where a "building" is treated as a pickable object, even though this is not realistic. As a result, while our framework can identify many domain-specific violations, it may not capture all real-world feasibility constraints—particularly for tasks beyond those modeled in RoboEval. We will clarify these points further in the revised limitations section.
>
> > The authors did not commit to releasing their code. Although the appendix provides details, it would be rather difficult to reproduce this system
>
> Yes, we will release a project website with implementation code, generated datasets, and checkpointed models.

---

> > ### Comment · Reviewer_LLwZ · 2025-06-08
> >
> > Thank you for your clarifications. I maintain my rating.

---

### Official Review · Reviewer_PKmn · 2025-05-14

**Rating:** 7
**Confidence:** 3
**Ethics Flag:** 1

**Summary:**

This work proposes a method to generate *grounded* synthetic training data for service robots.

A naive approach to generate novel tasks, e.g., using self-instruct, results in tasks that aren't solvable or compatible with the capabilities of models, and constraints and initial state of the environment. To address this issue, authors develop an approach to generate task-specific environments on the fly.

In particular, they assume executable robot APIs. They few-shot prompt LLMs to generate pairs of instructions and solution codes. They then invoke a type and constraint satisfaction-based approach to initialize a compatible starting state for the environment. If it doesn't satisfy, they reject the sample, otherwise, they find more initial states that pass the solution to ensure it is robust. Finally, once the solution has been established, they regenerate the instruction to "align" the instruction and solution.

Using this approach, they generate training data using RoboEval APIs, and show that training on this data leads to substantial improvements for small open models (more so than the baseline of just plain self-instruct and evol-instruct).

**Questions To Authors:**

- Have you tried using different/larger model for data generation? If so, do you see different/improved downstream results?
- Are comparisons with baseline data synthetis methods, OpenInstruct and EvolInstruct, size normalized? I.e., all three methods use the same sized training data?
- Have you tried training models larger than 8B? Using models under 8B is reasonable given your on-device deployment motivation. But I am still curious if these improvements also hold up for training larger models.

**Reasons To Accept:**

- The improvements over baselines are consistent and substantial.
- A novel and scalable approach to generate robot training environments on the fly.
- The paper is clearly written and suitable for this conference.

**Reasons To Reject:**

The experiments are shown on only one dataset, RoboEval. It's unclear if the results will hold up elsewhere.

I should note, though, although I am quite familiar with Code LLMs and their evaluations, I am not much familiar with robot evaluation benchmarks. So feel free to correct me if you think there's a good reason to choose only one here.

---

> ### Author Response · Authors · 2025-06-01
> **Response**
>
> We thank the reviewer for their constructive feedback. We will address the points raised below.
>
> > The experiments are shown on only one dataset, RoboEval. It's unclear if the results will hold up elsewhere. I should note, though, although I am quite familiar with Code LLMs and their evaluations, I am not much familiar with robot evaluation benchmarks. So feel free to correct me if you think there's a good reason to choose only one here.
>
> While evaluating general-purpose code generation is well-studied, robot program evaluation presents additional challenges due to its embodied nature—success requires not just correct code execution but also that the program respects real-world physical and task-specific constraints. These constraints are often specific to the tasks the robot performs and its physical embodiment (e.g., RoboEval focuses on mobile service tasks such as tour guiding, finding items, delivering mails, etc). To our knowledge, there are currently no other readily available benchmarks that provide both a set of robot APIs and a means to test LLMs’ ability to generate robot programs. We would also like to emphasize that verifying program correctness with respect to real-world and embodied constraints remains under-explored in robot code generation, which was a key motivation for our work.
>
> > Have you tried training models larger than 8B? Using models under 8B is reasonable given your on-device deployment motivation. But I am still curious if these improvements also hold up for training larger models. Have you tried using different/larger model for data generation? If so, do you see different/improved downstream results?
>
> While we recognize the importance of exploring larger models for data generation, we were constrained by available compute resources. In designing our experiments, we prioritized two key dimensions: comparing different base models and different baseline algorithms. Exhaustively evaluating combinations along these axes already fully utilized our compute budget. As a result, we did not experiment with larger models, which would require significantly more resources. We agree this is a valuable direction for future work and hope to investigate it as resources allow.
>
> > Are comparisons with baseline data synthesis methods, OpenInstruct and EvolInstruct, size normalized? I.e., all three methods use the same sized training data?
>
> Yes, all comparisons with baseline data synthesis methods are size-normalized; all three methods use the same amount of training data to ensure a fair and meaningful comparison.

---

### Author Response · Authors · 2025-06-01
**General Response**

We thank the reviewers for their thorough and insightful feedback. We are encouraged by the strong consensus in favor of accepting our work, noting that it (1) introduces a novel, scalable framework for verifying robot programs by synthesizing simulation environments on-the-fly and leveraging program-analysis techniques (highlighted by Reviewer PKmn & Reviewer txhX); (2) consistently and substantially outperforms baselines (noted by Reviewer PKmn & Reviewer LLwZ); and (3) is clearly presented—from the main text and figures to the detailed appendices—making the methodology easy to follow (recognized by all reviewers). Under the individual reviews, we will address each reviewer’s concerns and questions in detail.

---

### Decision · Program_Chairs · 2025-07-08

**Decision:**

Accept

**Comment:**

Empiricism, Data, and Evaluation: Mixed/Acceptable.

Reviewers highlighted thorough experimental design and substantial improvements over baselines, albeit on only one dataset (RoboEval), raising questions about generalizability. The evaluation is rigorous within its scope but narrow in breadth.

Technological Impact: Great.

All reviewers recognized the novel, scalable framework for robot program verification as a valuable contribution. The authors committed to releasing code, datasets, and models. The approach introduces program analysis techniques to the LLM community in a practical way.

Ambition, Vision, Forward-outlook: Mixed/Acceptable.

One reviewer questioned the forward-looking value given that frontier models outperform the approach. In my opinion, even if the specific results fade in relevance, the suggested approach has more fundamental research value. The proposed task+code data generation approach may offer a solution to the instruction dilution problem that seriously limits applicability of frontier models to robot code generation. There is room to extend it to physical-based constraints, but that was clearly out of scope for the paper.

Understanding Depth, Principled Approach: Great.

Reviewers appreciated the principled application of program analysis to robot program verification. While one reviewer questioned the novelty of the program analysis itself, all agreed the adaptation to embodied AI represents a meaningful technical contribution.

Clarity, Honesty, and Trust: Great.

Universal praise from reviewers for clear presentation, writing, helpful figures, and detailed appendices. Authors were transparent about limitations and honest about resource constraints during the rebuttal.

Overall: The submission excels on 3 out of 5 criteria, and meets the bar on the other 2.